# Ecology and Evolutionary History of *Diabrotica* Beetles—Overview and Update

**DOI:** 10.3390/insects13020156

**Published:** 2022-01-31

**Authors:** Astrid Eben

**Affiliations:** Julius Kühn–Institut (JKI), Federal Research Centre for Cultivated Plants, Institute for Plant Protection in Fruit Crops and Viticulture, Schwabenheimer Straße 101, 69221 Dossenheim, Germany; astrid.eben@julius-kuehn.de; Tel.: +49-(0)-394647-4760; Fax: +49-(0)-394647-4805

**Keywords:** antibiosis, cucurbitacins, *Diabrotica*, corn, coevolution, entomopathogenic nematodes, teosinte

## Abstract

**Simple Summary:**

This review provides an overview on selected aspects of the ecology and biology of *Diabrotica* leaf beetles. A special focus is on the western corn rootworm as a major pest insect on corn. Furthermore, general information on host plant relationships and natural enemies of this beetle group is presented. Current knowledge of these leaf beetles is mostly focused on a limited number of economically important species. For the majority of the species in the group little to nothing is known about their biology and their host plants. This information, however, could be useful for future plant breeding programs and pest control strategies.

**Abstract:**

An overview is given on several aspects of evolutionary history, ecology, host plant use, and pharmacophagy of *Diabrotica* spp. with a focus on the evolution of host plant breadth and effects of plant compounds on natural enemies used for biocontrol of pest species in the group. Recent studies on each aspect are discussed, latest publications on taxonomic grouping of *Diabrotica* spp., and new findings on variations in the susceptibility of corn varieties to root feeding beetle larvae are presented. The further need for in-depth research on biology and ecology of the large number of non-pest species in the genus is pointed out.

## 1. Introduction

Leaf beetles of the section Diabroticites (Chrysomelidae: Galerucinae: Luperini) comprise 823 species. Among these, the most species rich genus is *Diabrotica,* with about 400 described species [1,2,3]. The majority of species (*n* = 354) are grouped in the *fucata* group, and these species are supposed to be polyphagous on a number of plant families and multivoltine with more than one generation per year, while fewer of the *Diabrotica* spp. are oligophagous and univoltine: 24 species in the *virgifera* group and 11 in the *signifera* group [4]. Species of the genus *Diabrotica* are native to North and South America [2], with the greatest diversity found in Mexico and in Brazil. The highest number of species is found in Central America, where the group has its evolutionary origin [5,6]. The species of the *signifera* group are distributed only in South America [2,6].

Reliable information on biology, ecology, and above all host plant relationships is scarce or unavailable for many species [7]. To date, most data were published on the few species that are economically important as pest species—namely, *Diabrotica balteata* LeConte, *D. barberi* Smith & Lawrence, *D. undecimpunctata howardi* Barber, *D. virgifera* LeConte, and *D. speciosa* Germar [2,8,9].

In North America, *D. virgifera*, or western corn rootworm, is the most important pest insect on corn in the USA [10,11]. In the 1990s, this species was accidentally brought to Serbia, in Eastern Europe, where it quickly spread over large parts of eastern and central Europe and became a threat to the corn-growing regions foremost in Hungary and Germany [12,13,14,15,16]. Nevertheless, in those newly invaded regions, cultivating corn in a two- or three-year rotation with non-monocotyledon crops made it possible to maintain the populations of *D. v. virgifera* below economic threshold level [17,18]. *D. v. virgifera* is found in most European countries and was eradicated in Belgium, Netherlands, and United Kingdom [19], whereas in France and Germany, for example, populations are maintained below levels of economic damage due to intensive monitoring efforts and corn rotation strategies with particular regulations for each susceptible region and country [18].

## 2. Life Cycle

Species in the genus *Diabrotica* feed as adults on leaves and pollen of their host plants; one female can deposit 300–400 eggs in the soil at a depth of about 5–15 cm close to the roots of host plants, and larvae develop while feeding and tunnelling in host plant roots. Larval development encompasses three instars and mature larvae pupate in a soil chamber constructed by the late third instar larva. The pupal state lasts 7–10 days [2]. Newly hatched adults are softbodied and light coloured with scarcely visible elytral patterns (Eben, pers. observ). The entire life cycle of multivoltine species is completed within about 30 days. Moreover, univoltine species overwinter as eggs, and the multivoltine species spend the dry and cold season often as diapausing adults [2,20]. In *D. barberi*, an important pest species on corn, it is known that the egg diapause can be extended to two years due to availability of corn host plants in a biannual corn–soybean crop rotation system in Nebraska [21], whereas in semi-tropical and tropical climates, such as in Brazil, a multivoltine species such as *D. speciosa* can develop up to six generations per year [9].

## 3. Host Plant Use—Plant Resistance

*Diabrotica* species feed on plants from about 50 different families [6]. Some species are polyphagous—for example, a number of species in the *fucata* group—while others show narrow host use—i.e., most species in the *virgifera* group [8]. Information on host plants, however, remains scarce and difficult to find for most of the Diabroticites that are not of economic importance [7,22]. *Diabrotica* spp. were described to feed on at least 60 different crop plant species, but only a small number of species can be considered pests of these host plants [23,24]. Because some species distributed in the USA are of high economic importance as pest insects on corn, soybean, squash, and beans, their life cycle, their host plants, and their phylogenetic relationships have been well-studied [8,25,26].

Polyphagous species in the *fucata* group of the genus *Diabrotica* and their respective host plants are mainly found in the tropics and sub-tropics. Their distribution and abundance is thus often patchy and restricted to small areas. These habitats are very much in contrast with those used by prairie inhabiting and corn feeding species of the *virgifera* group, where host plants grow in large, landscape scale monocultures [7,27,28,29]. For many species, however, the knowledge of non-cultivated host plants remains restricted or unknown, even more so for larval hosts. The evolutionary switch of host plant families and the broadening or reduction in the host range in some species probably occurred in parallel with speciation events within the genus *Diabrotica* since several hosts were independently lost and gained throughout the evolution of the genus.

The intimate relationship of some species in the *virgifera* group with monocotyledonous plants in North American prairie habitats, mainly with corn, was long supposed to have originated in southern Mexico or further south in Guatemala [2,30]. Recently, it was confirmed that wild Mexican Balsas teosinte (*Zea mays* ssp. *Parviglumis diploperennis* Iltis and Doebley) is the ancestor of the modern, domesticated maize [31]. Domestication of maize in Mexico occurred about 9000 years ago [31,32], whereas the beginning of the diversification of cucurbits was dated back to at least 50 M years ago [33], with Asia being the area of origin of the family Cucurbitacae. The domestication of the genus *Cucurbita* was dated to about 10,000 years ago and is supposed to have taken place in northeastern Mexico [5,34,35].

Unfortunately, teosinte species in their native habitat are threatened by extinction before being investigated in detail for a number of potentially important characteristics, i.e., resistance towards biotic and abiotic stress. For example, one study of resistance mechanisms in these wild corn ancestors provided evidence for greater resistance against herbivores in teosinte than in domesticated corn [36]. Like a number of other herbivores, *Diabrotica* spp. feed on shoots and on roots of corn. The damage caused on the belowground root system may cause plants to partially fall to the ground and possibly wither or dry out. In contrast with corn, teosinte is a perennial plant and provides food for more than one generation of rootworms. Nevertheless, during comparative field investigations in Mexico, rootworms were found more abundant on corn than on teosinte [37]. This observation might be due to the generally much higher biomass in both, below- and aboveground plant parts of cultivated corn plants when compared with less sumptuous teosinte plants. Maize is well studied with regard to its inducible and constitutive defences such as hormones, volatile compounds, peptides, enzymes, and physical defences. Much less is known about the defence system of teosinte. Erb et al. [38] found that the response of corn and teosinte to simultaneous root herbivory by *D. v. virgifera* and *Spodoptera frugiperda* (J.E. Smith) (Lepidoptera: Noctuidae) was more intense for teosinte. In general, wild Mexican land races and teosinte seem to be more tolerant of herbivores than domesticated corn. Furthermore, since corn rootworms are associated with several symbiontic gut microbes, this interaction might affect plant defences as well as insect physiology [36,39,40,41].

Volatile compounds emitted by plants after herbivore attack are also broadly studied for corn [42,43]. These compounds are induced through feeding by rootworm larvae and reported to be highly variable in the different maize varieties [44]. The volatiles interact with above and below ground natural enemies of herbivores, such as parasitic wasps and entomopathogenic nematodes, respectively [45]. The capacity for production of induced defences was reported to be reduced or partially lost in North American domesticated maize varieties but was still broadly found in teosinte [46]. In a laboratory study, maize resistance to root herbivory by *D. v. virgifera* was compared for several biotypes of Balsas teosinte, Mexican landraces, US landraces, and US inbred corn [47]. The results showed that larval weight and survival was highest on US landraces and lowest on teosinte. The authors interpreted their findings as evidence for decreased corn resistance to feeding damage by rootworm larvae through domestication (landraces). Nevertheless, this negative correlation was only partially mediated through breeding since US inbred lines of maize were more resistant than US landrace ancestors. Such a negative correlation for domestication and resistance was also found for Mexican landraces that were less resistant to feeding damage than their teosinte ancestors. A similar study with Brazilian landraces and cultivars of maize found higher level of resistance towards feeding by *D. speciosa*, the most important Diabroticite pest species in Brazil, in two genotypes [48]. Larval development was longer and mortality was increased on one landrace 1 of 17 tested, and on one of two cultivars of maize. In laboratory experiments, those varieties were found to produce higher levels of lignin, cellulose, and fibres in the roots for a prolonged period after larval feeding. The identification of maize-genotypes with such so-called “antibiosis” effects on rootworms are of importance for future breeding programmes of rootworm-resistant corn varieties or for their use in crop rotation strategies.

## 4. Evolutionary History

The large diversity of species in this beetle group has been interpreted in different ways based on assumptions of adaptive radiation, coevolution [49], and competitive exclusion plus geomorphological patterns [7,50]. The origin of the Neotropical genus *Diabrotica* can be dated back to the Cretaceous, and its diversification began about 60 Mya with a peak between 30 and 45 Mya [7]. This is much earlier than previously proposed by Metcalf and Lampman [51], who related the diversification in Diabroticite beetles to the agricultural practice of intercropping corn with beans and cucurbits all over the Americas (inter-cropping theory) [34]. In Mexico, the cultivation of maize from its ancestor, teosinte (*Zea mexicana*), began about 9000 years before the present in the southern regions of Mexico [31], from where its use in agriculture spread north to Arizona, New Mexico, Utah, and Colorado (for more details see [10]). This is much later than the speciation events in *Diabrotica.*

Moreover, the diversification of *Diabrotica* spp. cannot be interpreted without taking into account its intricate relationship with wild species of the family Cucurbitaceae [2,5]. This plant family is characterized by the production of bitter, toxic secondary compounds—the cucurbitacins that render the plants unpalatable for most herbivores, including humans [35,52,53]. To date over 40 cucurbitacins and cucurbitacin-metabolites are known from Cucurbitaceae, and many of them have important medicinal properties [54]. The tetracyclic triterpenoids are typically found in wild species of cucurbits [55] and act as compulsive feeding arrestants for *Diabrotica* species [56,57]. When Diabroticite beetles encounter plant tissues with cucurbitacins they are arrested and continue to feed compulsively on the bitter tissues (Figure 1) [58,59,60,61,62].

In a number of cucurbits, the highest concentrations of cucurbitacins are found in cotyledons, roots, and seeds of cucurbit species [63]. The beetles even abandon suitable host plants and prefer cucurbitacin containing tissues [60]. This compulsive feeding behaviour results from contact of receptors on the maxillae with the bitter cucurbitacins [64,65]. This trait might indicate an ancestral relationship with Cucurbitaceae that is maintained even in species no longer associated with cucurbits or in species where larvae develop on non-cucurbit hosts [2,60,66,67], probably due to benefits obtained through sequestration of the bitter compounds [68]. By feeding on cucurbitacin containing plant tissues, the beetles sequester these compounds in their haemolymph [51]. This was interpreted as an example of chemically mediated coevolution, as well as for the so-called feeding behaviour named pharmacophagy [69,70]. The latter is based on the fact that the beetles not only prefer to consume the bitter cucurbit tissues but also sequester the imbittering secondary compounds. Consequently, adult and larval *Diabrotica* spp. are protected from natural enemies by those sequestered plant compounds [68,71,72,73]. However, results on chemical defence are contradictory and depend on which beetle species and natural enemy are studied (see details in [74]). Furthermore, cucurbitacins are transferred into eggs and larvae through female beetles [68] and can act as antibiotics against the entomopathogenic fungus *Metarhizium anisopliae* [72].

Under field conditions in Veracruz, Mexico, mostly male beetles of Diabroticite species accumulate in the flowers and on tissues of wild cucurbit species (Figure 2) [73].

Such a behaviour seemingly, what is called “lek formation”, might contribute to sequestration of cucurbitacins in male tissues as well as in sperm transferred to females during copulation. In this way the cucurbitacins that are transferred to the females might have been already detoxified in the males and formed conjugates which reduce the metabolic costs for the receiving females. Such possible mating advantage or female choice preference for more “bitter” males was tested by Tallamy et al. [75]. In their study, however, this hypothesis could not be proven. Moreover, the compulsive feeding upon encounter of cucurbitacins was successfully incorporated in pest control strategies against the western corn rootworm. There, cucurbitacins were offered in combination with insecticides [76,77].

Cucurbitacin metabolism and detoxification in *Diabrotica* spp. differ in species with more restricted host plant breadth, such as *D. virgifera virgifera* compared with polyphagous *D. balteata* or *D. undecimpunctata howardi*. The least metabolic costs were reported from specialized cucurbit feeders such as closely related *Acalymma* spp. [53]. Among Diabroticite beetles, all 73 known species in the genus *Acalymma* [1,78] are specialized on cucurbits as adults and larvae. One species, *A. vittatum* (F.), is an important pest species on squash in the USA [79,80]. Like most investigated Diabroticites, these species show compulsive feeding on cucurbitacin-laden plant tissues and prefer them over non-cucurbitacin-containing host plants [61,62,81]. In a new study [82], the importance of cucurbitacin type and concentration in *Cucurbita pepo* cultivars was investigated. This research found that adult *A. vittatum* host preference was positively correlated with cucurbitacin content of cotyledons, but preference was also affected by the presence of other chemical compounds in the first true leaves that contained very low cucurbitacin levels. Moreover, cucurbitacin content in the cotyledons was not increased, i.e., induced, through feeding. The results of the genetic mapping and gene expression analyses were interpreted as an indication that genes responsible for the production of cucurbitacins in cotyledons and roots might contribute to intraspecific differences in plant susceptibility to herbivores. Such a finding is important for future breeding programs of edible Cucurbitaceae species.

## 5. Taxonomy and Phylogenetic Relationships

The taxonomic positioning and the relationships among species in the genus *Diabrotica* have been examined recently [3,83]. The authors described 18 new species for North and Central America and extended the taxonomic key including more morphological characters to distinguish species of the genus *Diabrotica* from other Diabroticite species. However, the revision was restricted to the North and Central American *fucata* and *virgifera* species groups with previously 123 reported species [1]. Only 107 species remained as valid species in the genus with a distribution in North and Central America. Unfortunately, only scarce data on host plants were provided in this new revision, probably due to the lack of information on sampling locations of the museum specimens. Especially, the species-rich and less-studied South American *fucata* group and *signifera* group still warrant exhaustive research on biology, ecology, host plant associations, and phylogeny.

An interesting morphological parameter used for species comparisons and taxonomic grouping is the sexual dimorphism of the basi-tarsus on the pro- and mesothoraxic legs found in *D. collicola* Cabrera & Cabrera Walsh [84], *D. speciosa*, *D. viridula* (Fabricius) [85], and other species in the tribes Diabroticites and Phyllectrites [86,87]. The adhesive setae found in males of some species were supposed to play a role in copulatory behaviour and male–male competition [88]. This morphological character can distinguish between male and female specimens in a non-destructive way and can also be used with living individuals.

Among the species comprising the genus *Diabrotica*, some are specialists on cucurbits (i.e., *D. scutellata*), whereas others are extremely polyphagous (*D. balteata*, *D. speciosa*). Since larvae are hidden from sight as below ground root-feeding and pollen-feeding adults are often found on hosts from diverse plant families, the evolution of host plant use in the genus cannot yet be fully understood. Efforts are further complicated by the lack of a complete phylogeny for this Neotropical genus. A phylogenetic reconstruction including gene sequences of 40 species found monophagous feeding behaviour as the ancestral trait [89]. The evolutionary diversification in the genus *Diabrotica* was initiated with the inclusion of host plant families that led to polyphagy. Secondary regressions to oligophagy and monophagy on cucurbits later occurred in some species. In the cited study, data on host plant use were obtained based on field observations and published host records from the literature. Overall, a high plasticity in the genus *Diabrotica* with regard to the evolution of host plant breadth was found. One characteristic that facilitated such an evolutionary pathway might be the subsequent optimization of cucurbitacin metabolism and sensory reception [59,90]. Further studies of cucurbitacin detoxification in non-pest species as well as their cucurbitacin receptor physiology and biochemistry are needed.

## 6. Natural Enemies of Diabroticites—Old and New Findings

Long common evolutionary histories in the native habitat made natural enemies adapted to the behavioural and chemical defences of their hosts. Parasitic flies of the species *Celatoria compressa* Wulp (Diptera: Tachinidae) were reared from several species of Diabroticite beetles in Veracruz, Mexico [71]. Based on extensive field surveys in central Mexico [73], this fly species was later considered for classical biological control of *D. v. virgifera* in Europe [91,92]. The fly, however, cannot be considered a specialist on species in the genus *Diabrotica*. Therefore, mass releases in the newly invaded areas can be undertaken only after intensive investigation taking into account potential negative effects on native Chrysomelid species. In Mexico, *C. compressa* is not abundant—like most parasitic insects, we found only around 5–10% of adult Diabroticites parasitized by the flies during several years of intensive field surveys [71,73].

With a focus on the most damaging life stage, the root-feeding larvae of *Diabrotica* spp., soil-inhabiting entomopathogenic nematodes of the families Heterorhabditidae and Steinernematidae were broadly investigated [39]. A significant effect of larval host plants on the efficiency and propagation of nematodes was found for *D. undecimpunctata howardi* and its main hosts. Feeding on the roots of corn and peanuts was positively correlated with greater rootworm weight and increased survival, whereas larvae fed on cucurbit roots were smaller and showed higher mortality [40]. Most importantly, it was the symbiontic bacteria in the nematodes that were negatively affected by cucurbitacins in roots of domesticated *C. pepo* [41,93]. In this sense, not only cucurbitacins can affect these symbionts but also secondary compounds produced by maize. Research on metabolites sequestered by *D. v. virgifera* larvae when feeding on maize roots found benzoxazinoid glucosides. Rootworm larvae stabilize benzoxazinoid derived from corn roots by glucosylation. When infested by entomopathogenic nematodes, the larvae hydrolyse the compound and release toxic MBOA for their defence [94]. In this tri-trophic level system, the toxic compounds act against the 4th trophic level, the symbiontic bacteria in the nematodes. Recently, Bruno et al. [95] detected that most of 40 Mexican isolates of *Heterorhabditis bacteriophora* were resistant to benzoxazinoids, secondary compounds in the tested maize varieties. These compounds were sequestered by *D. v. virgifera* larvae and thus acted as chemical defence against some of the nematode isolates. Interestingly, all nematode isolates successfully infested polyphagous *D. balteata* larvae that did not sequester these metabolites from corn roots. Nevertheless, most of the nematodes obtained from the soil of Mexican cornfields were resistant to maize metabolites encountered in their insect hosts and effectively killed oligophagous *D. v. virgifera* larvae. Based on those results, it is unknown if symbiontic bacteria in entomopathogenic nematodes are able to detoxify the plant metabolites encountered in their host and make larval hosts suitable for their own propagation. Furthermore, the highly variable nutritional quality of the different maize genotypes tested might also affect susceptibility of both larvae and nematodes to each other.

## 7. Conclusions

Despite the economic importance of several *Diabrotica* spp. and their well-studied feeding relationship with corn and bitter plants of the Cucurbitaceae, knowledge of their biology related to topics such as diapause requirements or voltinism, is still unavailable for most species. Little information on host plant use of adults and larvae is published for the majority of the Central and South American species that are not of economic interest. Taxonomic studies, further phylogenetic reconstructions, and much broader knowledge of host plant associations and the distributions of the many species in Mexico and Central and South America are needed. That information will help explain why some species develop into such successful pests, such as the western corn rootworm, and how pest species might be controlled with environmentally sustainable means in the near future.

## Figures and Tables

**Figure 1 insects-13-00156-f001:**
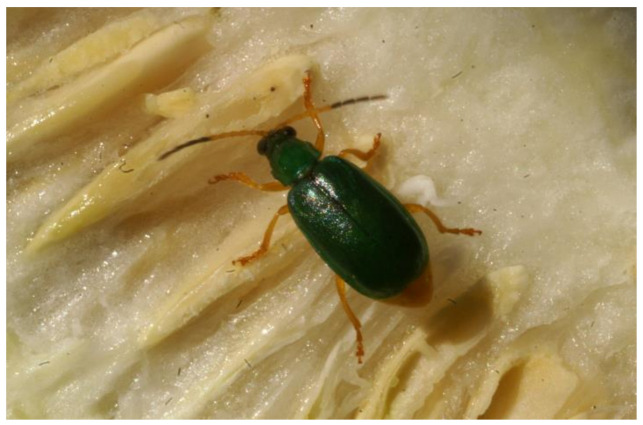
Diabroticite beetle of the species **Diabrotica* dissimilis* Jacoby feeding on the seeds of an open fruit from wild, bitter *Cucurbita okeechobeensis* ssp. *martinezii* (L.H. Bailey). This plant species of the family Cucurbitaceae is native to Veracruz, Mexico (Photo: A. Eben).

**Figure 2 insects-13-00156-f002:**
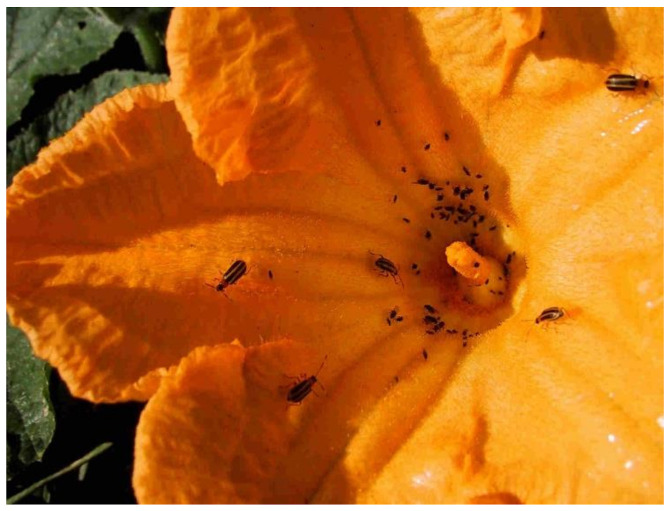
Flower of *Cucurbita pepo* L. with several Diabroticite beetles of the species *Acalymma blomorum* Munroe & Smith 1980 in the surroundings of Coatepec, Ver., Mexico (Photo: A. Eben).

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
