# Peer review of "Ecology and Evolutionary History of Diabrotica Beetles—Overview and Update"

_insects, 2022, doi:10.3390/insects13020156_

Round 1

Reviewer 1 Report

Overall, the review by Astrid Eben will likely become acceptable after modifications.  I was a bit disappointed in quality of information and writing. There are numerous typing errors and several factual errors.  For instance, the author states that larvae of the genus Diabrotica have four instars, which certainly is in error for those species in USA. From my perspective, the review marginally meets its stated goal and could be accepted after fixing errors and omissions.

Line 21 – citing papers that are 40 and more than 50 years old for how many species are in the genus Diabrotica.  Nothing since then?

Line 31 – change ‘scare’ to ‘scarce’

Line 33 – add Diabrotica barberi to this list.  There is more published on this species (154 in Scopus) than balteata (74 in Scopus) or speciosa (102 in Scopus).

Line 34 – do not capitalize the ‘w’ in western, the ‘c’ in corn or ‘r’ in rootworm.

Line 66 – awkward transition and differentiation from the above paragraph

Line 67 – I would have expected the book review chapter by Tallamy to be cited here.  It is more comprehensive that the manuscripts cited.

Tallamy, D.W., B.E. Hibbard, T.L. Clark, and J.J. Gillespie.  2005.  Western corn cootworm, cucurbits, and cucurbitacins, pp. 67-93.  In Western Corn Rootworm: Ecology and Management.  Stefan Vidal, Ulrich Kuhlmann, and Richard Edwards (eds.), CABI publishers, Wallingford, United Kingdom.

Line 91 – not sure of final format, but if using American English it should be “lek formation”, not ,,lek formation”.

Line 98 – do not capitalize the ‘w’ in western, the ‘c’ in corn or ‘r’ in rootworm.

Lines 105-6 – there are 118 manuscripts with a simple search of Acalymma vittatum.  The cited manuscripts do not seem the best available to support the statement.

Line 106 – change ‘shows’ to ‘show’

Line 120 – so, are lines 20 and  21 correct or incorrect, since those figures cite older literature?

Line 123 – capitalize the ‘T’ in ‘the’ at the beginning of the sentence

Line 123 – pest species in the USA have only 3 instars.  This paragraph speaks to all Diabrotica, so is in error and must be fixed.

Line 189 – the author states that information on maize ancestor was recently documented and then cites a 20-year-old reference.  There are dozens of more recent citations on this overall point. 

Line 205 – change ‘upground’ to ‘above ground’

Line 288 – do not capitalize the ‘w’ in western, the ‘c’ in corn or ‘r’ in rootworm.

Line 289 – same parentheses issue as in line 91 above.

Author Response

Dear Reviewer,

thank you very much for your helpful comments. I addressed all of them and made the corresponding changes. Please see my answers in capital letters in the file attached. 

Reviewer 2 Report

The manuscript presents the (Ecology and evolutionary history of Diabrotica beetles). The manuscript might have some merit in terms of work done

But, the paper is poorly written and has many grammatical errors.

The data is not well displayed. Why the author didn’t use some table, figure..etc) ?.

Some comments are reported.

34 – In the introduction part, the author focuses only on D. virgifera virgifera? What about other species situations?

40 – Which species?

50 -Please add (million years ago (Mya)).

51-57 re-write.

61 – Add references

62-65 not clear

89-90 which species?

92 - Add references

94 - Add references

96 - Replace to (could).

123- The.

155 Life cycle… (is very brief). Need more details.  

156-159 Add references

254 - Add references

Author Response

Dear Reviewer,

thank you for your comments and suggestions. I addressed them and made the corresponding changes. Please see my answers in capital letters in the file attached.

Round 2

Reviewer 1 Report

The author has adequately addressed concerns.  This review is fine for publicaiton.

Author Response

Thank you again for your helpful suggestions and corrections during the first revision round. I am glad to read that you regard the revised manuscript as acceptable for publication. 

Reviewer 2 Report

The manuscript could be accepted in its current form

Just re-check the manuscript style for example ( line 31 you should add ( [ ).

Author Response

Dear reviewer,

thank you again for your helpful comments and suggestions during the first revision. Missing bracket in line 31 has been added.

I am glad to read that you consider this revised version of my review as acceptable for publication.